# Chronicling the Journey of Pneumococcal Conjugate Vaccine Introduction in India

**DOI:** 10.3390/vaccines13040432

**Published:** 2025-04-21

**Authors:** Pawan Kumar, Arindam Ray, Amrita Kumari, Abida Sultana, Rhythm Hora, Kapil Singh, Rashmi Mehra, Amanjot Kaur, Seema Singh Koshal, Syed F. Quadri, Shyam Kumar Singh, Arup Deb Roy

**Affiliations:** 1Immunization Division, Ministry of Health & Family Welfare, Government of India, New Delhi 110011, India; drpawan.mohfw@gmail.com (P.K.); kksingh@unicef.org (K.S.); 2Gates Foundation, New Delhi 110067, India; arindam.ray@gatesfoundation.org (A.R.); amrita.kumari@gatesfoundation.org (A.K.); 3John Snow India, New Delhi 110070, India; rhythm_hora@in.jsi.com (R.H.); rashmi_mehra@in.jsi.com (R.M.); amanjot_kaur@in.jsi.com (A.K.); seema_koshal@in.jsi.com (S.S.K.); syed_quadri@in.jsi.com (S.F.Q.); shyam_singh@in.jsi.com (S.K.S.); arup_debroy@in.jsi.com (A.D.R.)

**Keywords:** pneumococcal conjugate vaccine, COVID-19, universal immunization program

## Abstract

Background: Globally, pneumonia claims the lives of about 700,000 children under the age of 5 every year. Pneumococcal conjugate vaccine (PCV) was introduced in India phase-wise, beginning in high-burden states, and the rollout was completed nationwide by 2021—representing a major initiative by the Ministry of Health and Family Welfare (MoHFW). Despite the challenges posed by the COVID-19 pandemic, the campaign succeeded in maintaining progress and achieving nationwide coverage. This narrative review highlights the significant decisions, processes, and coordinated efforts of the various stakeholders involved that led to this successful PCV rollout. Methodology: A comprehensive desk review of both published and unpublished literature relevant to pneumonia burden and the efficacy and effectiveness of PCVs, along with documentation of PCV introduction and the scale-up was carried out. Results: The documentation of the PCV journey has been broken down into four sections: pre-introduction, PCV Phase-I introduction, pan-India rapid expansion, and the period post-introduction. Since the nationwide rollout in 2021, PCV coverage in India has steadily increased, reflecting successful immunization efforts. WUENIC, which is an annual WHO, and UNICEF estimates of national immunization coverage also show a positive trend in vaccination coverage (PCV booster coverage = 25% (2021), rising to 83% (2023), aligning with the goals of the WHO and UNICEF’s Global Action Plan for the Prevention and Control of Pneumonia and Diarrhoea (GAPPD). Conclusions: The phased rollout was an ambitious effort by the MoHFW, which was particularly challenging given the overlap with the COVID-19 pandemic. Despite these hurdles, the MoHFW, along with strong collaboration from development partners and stakeholders, successfully navigated the complex rollout. Future studies on the role of PCVs in reducing antibiotic resistance and the economic benefits of PCV introduction could help policymakers sustain funding and prioritize vaccine procurement decisions.

## 1. Introduction

Pneumonia is the leading cause of morbidity and mortality in children under the age of 5, causing the deaths of about 700,000 children aged under 5 every year, or about 2000 every day [1,2]. According to the Cause of Death Statistics report of the Registrar General of India 2017–2019, pneumonia emerged as one of the main causes of death for children under 5, accounting for 17.5% of fatalities in India [3]. A number of studies have brought to the fore that Streptococcus pneumoniae is the leading cause of pneumonia and it also causes a wide spectrum of diseases, including both fatal invasive (bacterial meningitis and sepsis) and non-invasive diseases (otitis media and sinusitis) in children under 5 [4,5,6].

Considering the large burden of disease and the vaccine-preventable nature of pneumococcal diseases, the World Health Organization (WHO) (2012) released a position paper on pneumococcal conjugate vaccines (PCV10 and PCV13), endorsing them as a safe and effective tool to prevent pneumococcal diseases [7]. In India, the National Technical Advisory Group on Immunization (NTAGI) suggested the introduction of PCVs in the Universal Immunization Program (UIP) in 2015 [8]. Following this, the MoHFW of the Government of India (GoI) decided to introduce PCVs under the UIP in 2017 [9].

India embarked on its journey to introduce PCVs in a phased manner, commencing with five high-burden states in May 2017 [10]. This first phase included the following states: Himachal Pradesh, Uttar Pradesh, Bihar, Rajasthan, and Madhya Pradesh. Later, in 2021, the MoHFW announced a rapid pan-India expansion of the PCV rollout [11]. Introducing a new vaccine was a comprehensive task involving multiple processes spread over 4 years, multiple phases, and 36 states/UTs. This became especially challenging during the pan-India expansion phase when the country was struck by the second wave of the COVID-19 pandemic [12]. The unprecedented challenges, such as personnel travel restrictions and social distancing, posed by the pandemic made the nationwide scale-up more difficult, necessitating the development of various innovative tools, approaches, and processes to successfully roll out the vaccine.

The present study aims to chronicle the journey of PCV introduction in India from its inception to its pan-India expansion, along with a brief overview of the impact of the introduction and future research implications.

## 2. Methodology

The present study deploys a comprehensive desk review of the available published and unpublished literature to document the journey of PCV introduction in India. The review was developed in two stages. The first stage comprised a comprehensive literature search conducted on PubMed, the Cochrane database, and Google Scholar to identify literature published since its inception to November 2024. Literature covering topics related to the burden of pneumonia, pneumococcal pneumonia mortality in children under the age of 5, the efficacy and effectiveness of PCVs, and the documentation of PCV introduction and scale-up in India were considered. Gray literature included, but was not limited to, reports, newsletters, blogs, government documents, and speeches on PCV introduction available in the public domain. Government documents included the PCV rollout’s operational guidelines, meeting minutes, and training materials available in the public domain. The authors—including the National EPI Manager from the Ministry of Health & Family Welfare, donors from the Bill & Melinda Gates Foundation, and Program Managers and Program Officers from the lead technical agency, John Snow India—collaborated to share insights and reflect on their experiences with the introduction and expansion of PCVs in the second stage. The review was based on the literature studies and the authors’ opinions and experiences. To minimize the risk of bias, a systematic approach was employed, including a comprehensive literature search across multiple databases, the inclusion of grey literature to reduce publication bias, and the validation of findings through collaboration with diverse stakeholders from the government, donor agencies, and implementing partners. Instead of relying solely on predefined inclusion/exclusion criteria, expert judgment was applied to assess the relevance of the sources. This allowed for a more nuanced interpretation of the data rather than mechanical filtering.

## 3. The Journey

### 3.1. Pre-Introduction

#### 3.1.1. Evidence Generation and Synthesis

Pneumococcal conjugate vaccines (PCVs) were developed to overcome the shortcomings of capsular polysaccharide-based pneumococcal vaccines, which were less efficacious for children under the age of 2, the age when invasive pneumococcal disease (IPD) is most likely to cause death [13]. A number of national and international studies analyzing the effectiveness of PCVs helped generate evidence to support the introduction of PCVs in India.

Systematic reviews across high-, low-, and middle-income countries presented the effectiveness of all dosing schedules (2+1, 3+0, and 3+1) of PCVs at reducing invasive pneumococcal diseases in the under 5 years age group [14,15]. Another meta-analysis reported that the efficacy of PCVs in the reduction of invasive pneumococcal disease was 89%. Additionally, a PCV of any valency has shown its effectiveness against pneumonia, meningitis, bacteremia/sepsis, and otitis media [16,17]. Furthermore, a meta-analysis of impact studies brought to the fore that 3 years after the onset of a 2+1 dose schedule of PCV-7, with coverage of ≥ 80% throughout the ascertainment period, a 98% reduction in the incidence rates of IPD was reported [18]. Another study reported the effectiveness of PCV10 and PCV13 in reducing invasive pneumococcal disease and the early effects of bacterial meningitis in children under the age of 5 [19]. Additionally, a hospital-based surveillance network found that the serotypes covered under PCV13 caused 72% of bacterial meningitis [20]. Another study suggested that children exposed to PCV13 had a considerably lower chance of developing primary end-point pneumonia in cases of severe community-acquired pneumonia [21]. Several studies showcased the immunogenicity and safety of PCVs in the Indian population [22,23,24].

Thus, PCVs provide a dual benefit against childhood pneumonia and bacterial meningitis. There have been global reports of changing trends in the epidemiology of bacterial meningitis with the introduction of PCVs [25,26]. However, there are limited studies on such research in India.

#### 3.1.2. Decision-Making Process

The National Technical Advisory Group on Immunization (NTAGI) recommended the phased introduction of PCVs under the UIP in August 2015. This recommendation was based on study findings of the pneumococcal disease burden, the vaccine’s safety and efficacy, and the global experience, which were highlighted in the recommendations by the NTAGI [8,27]. The recommendations of the NTAGI were approved by the Empowered Program Committee (EPC) and subsequently approved by the Mission Steering Group (MSG) of the National Health Mission (NHM).

The National Pneumococcal Vaccine Expert Committee was constituted by the GoI to guide the introduction of pneumococcal vaccination in the country. The Committee recommended PCV13, a WHO-prequalified vaccine, as the preferred vaccine type for introduction in the UIP, based on the available information on product specifications and operational feasibility, including multi-dose presentation and compliance with an open vial policy [27]. For the introduction of PCV13 under the UIP, a dosing schedule of 2 primary doses at 6 weeks and 14 weeks, followed by a booster dose at 9 months, was recommended [28]. This dosing schedule was in line with the existing UIP schedule. Subsequently, in May 2016, the Expert Committee Group recommended the introduction of a PCV rollout program in the country under the UIP [27,29]. By providing vaccinations through the UIP, the Government of India guaranteed fair access to PCVs to the most vulnerable, the disadvantaged, and the underserved population of India; earlier, this access was restricted to the private sector [30].

### 3.2. PCV Phase I Introduction (2017–2020)

The PCV was initially introduced in 5 high burden states, namely, Himachal Pradesh, Bihar, Uttar Pradesh, Madhya Pradesh, and Rajasthan.

In its first year (2017) of introduction, the PCV was rolled out in 100% of the districts of Himachal Pradesh, 50% of the districts of Bihar, and 10% of the districts of Uttar Pradesh, covering a cohort of 7.8%. In 2018, the remaining 50% of districts in Bihar, the entire state of Madhya Pradesh, 25% of districts in Rajasthan, and an additional 20% of districts in Uttar Pradesh were included, covering 26.8% of the cohort. In 2019, 30.4% of the cohort was covered after the PCV was made available in 25% more districts in Rajasthan and 30% more districts in Uttar Pradesh. By 2020, the PCV was introduced in all the remaining districts of Rajasthan and Uttar Pradesh, making the PCV accessible to every child in these five states. Thus, by the middle of 2020, the PCV rollout was scaled up to the five high-burden states (Uttar Pradesh, Bihar, Madhya Pradesh, Rajasthan, and Himachal Pradesh), catering to 48.9% of the national cohort [10]. The geographic extent of PCV introduction from 2017 to 2020 is shown in Figure 1.

As recommended by the Expert Committee Group, the available PCV13 product was used in the selected geographies during this phase of introduction [27,29].

As with previous new vaccine introductions, the PCV was rolled out in a similar manner across the identified geographies, commencing with a comprehensive preparedness assessment, which is required at the national, state, and district levels prior to the introduction of a new vaccine. This was undertaken in a traditional paper-based format to ensure the smooth planning of processes [27].

This process was followed by cascade training, starting with a pool of national-level training of trainers (ToT), who provided state-, district-, and subdistrict-level training [31]. To establish a team of master trainers at the national level, a national ToT was held. It was attended by immunization partners, including officials from the identified states and partner organizations (WHO, UNICEF, UNDP, JSI, ITSU, and NCCVMRC). This was followed by in-person classroom training, from state to district to block (administrative unit) level. Detailed operational guidelines and frequently asked questions (FAQs) were developed and circulated in print among frontline health workers and program managers [32].

For demand generation, standard procedures of information, education, and communication (IEC) materials were developed and disseminated through mass media and social/digital media. Media sensitization workshops were organized at the state level to provide an overview of the universal immunization program, the pneumococcal disease burden, and the importance of PCVs [28].

As part of the preparation for PCV introduction in the selected states, it was ensured that sufficient cold-chain space was made available. Also, proper functioning of the cold chain is essential for transporting and storing vaccines under strict temperature control, without compromising their quality [28,32].

### 3.3. PCV Pan-India Rapid Expansion (2020–2021)

In 2019, Pneumosil, an indigenous vaccine developed by the Serum Institute of India that is a 10-valent PCV product, received pre-qualification from the World Health Organization, which deemed it safe and effective for use [33]. Consequently, Pneumosil was licensed for use in India in 2020, following which, it was launched as a PCV product by the then Union Health Minister on 28 December 2020 [31,34].

With the availability of a WHO-prequalified indigenous and affordable vaccine, the Government of India announced the nationwide expansion of the PCV rollout under the UIP in the 2021–2022 budget [11]. In response to the budget speech, the MoHFW undertook the pan-India PCV rollout’s rapid expansion in the remaining 31 states/UTs in 2021 [35]. It was intended that all 36 states/UTs would be covered, culminating in 27 million children having access to the PCV. Considering that the expansion coincided with the ongoing COVID-19 pandemic in 2021, several novel approaches were applied to each step of the new vaccine introduction, as mentioned below.

#### 3.3.1. Preparedness Assessment

Due to the unprecedented challenge put forth by the pandemic, the process of a traditional paper-based manual preparedness assessment, involving intensive documentation and supervisory visits, was not possible [36]. The lockdown-enforced travel and social restrictions necessitated an alternative approach to achieving this task [12]. As a result, an innovative, interactive, efficient, and user-friendly digital tool, the PCV Rollout Monitoring and Preparedness Tool (PROMPT) was developed to execute the process. This was a technology-driven solution created to carry out the preparedness assessment before the pan-India PCV expansion. This automated PROMPT tool was designed to eliminate the need for intensive manual documentation and reduce the time needed to complete the task. The interactive, user-friendly dashboard interface and real-time monitoring of assessment status made the preparedness assessment exercise more efficient, timebound, and trackable [36]. The preparedness assessment was conducted in all 31 states on time to match the stipulated timeline for PCV rollout expansion [32,37].

#### 3.3.2. Training

Since the service delivery of the entire UIP rested on the accredited social health activists (ASHAs) and auxiliary nurse-midwives (ANMs), the timely and effective training of these personnel is imperative. For the rapid scale-up across 31 states, a hybrid mode of training (a combination of online and classroom training) was utilized for all levels of health functionaries to overcome the disruption posed by the pandemic [12,38].

An online mode of training facilitated the training of a large cohort of health workers within a short span of time. FAQs and detailed operational guidelines were drafted and disseminated using soft copies, in addition to printouts, to overcome COVID-19 restrictions. This was an unprecedented process of training to be deployed, given the circumstances. However, to supplement the online training course, five animated videos based on the PCV FAQs were developed in 13 languages, including Hindi and English [39,40]. These were widely circulated among health workers through messaging platforms like WhatsApp. The translation of these videos into regional languages made communication, training, and understanding of the operational aspects of PCV introduction much simpler.

#### 3.3.3. Community Mobilization

Awareness generation measures were undertaken using traditional media (radio spots, posters, banners, hoardings, and leaflets), mass media (TV spots and newspaper advertisements), and social/digital media (WhatsApp videos and messages) [38,41]. The major regional electronic media outlets aired the live launch of the PCV, followed by a media workshop and a Q&A session with the media.

Celebrities, politicians, and other prominent public figures were engaged for community mobilization. Radio jingles were deployed for awareness generation about the availability of the PCV under the UIP. During the pandemic, a great deal of awareness was generated and imbued into the community about respiratory pathogens, hand hygiene, social distancing, and the importance of wearing masks, which helped create awareness and generate demand during PCV expansion through social media, mass media, and conventional methods, including flyers [12,42,43,44].

#### 3.3.4. Logistics and Cold-Chain Space Management

Before the pan-India expansion of the PCV, an assessment of cold chain space and equipment was conducted to ensure sufficient space for vaccine storage. This analysis of cold-chain space was performed as a part of the preparedness assessment to rule out any shortages in terms of space availability. Additionally, the augmentation of cold-chain space during the pandemic resulted in ample cold-chain space for PCVs. If there were non-functional deep freezers or ice-lined refrigerators (ILR), the State was notified, and the appliances were repaired or replaced with new ones [12,37]. Furthermore, a digital platform, the Electronic Vaccine Intelligence Network (eVIN), was leveraged throughout the country to monitor the vaccine supply chain, providing a real-time picture of vaccine logistics, including temperature monitoring, live stock positions, etc. [32].

#### 3.3.5. Monitoring and Supervision

Monitoring and supervision were required to track the status of program implementation, ensure accountability, and plan necessary corrective actions wherever needed. Following the pan-India PCV expansion, simultaneous rapid field monitoring was carried out to identify bottlenecks and provide feedback for immediate improvements. For this activity, standardized data collection formats were developed, covering all components of routine immunization. Rapid monitoring was undertaken by Government staff and an immunization partner at the block and session level. A mobile application was used to allow ease of data collection, collation, and visualization [45]. In 2021, PCV expansion to 31 states was completed within eight months of the budget announcement amidst the pandemic, demonstrating the strength and resilience of India’s healthcare delivery system [11]. This was a remarkable achievement for all stakeholders involved in the PCV’s introduction. Figure 2 depicts that since its launch in 2017, the PCV was expanded into all 735 districts in India, covering 100% of the population by 2021. Figure 3 shows the geographical representation of the PCV launch in 2017–2020 in a few states, with nationwide expansion in all the states by 2021.

Below is a table summarizing the adaptive measures employed during the COVID-19 pandemic to overcome challenges encountered during the PCV’s pan-India expansion.

### 3.4. Post-Introduction

#### 3.4.1. PCV Coverage

Since the pan-India expansion of the PCV in 2021, national-level PCV coverage has shown a steady upward trend. The steady increase in PCV booster (indicating the completion of the PCV dose schedule) coverage has been further corroborated in the recent WHO/UNICEF Estimates of National Immunization Coverage (WUENIC) [46]. Figure 4 illustrates a consistent upward trend for PCV booster coverage from 6% in 2018 to 25% in 2021 and a sharp rise to 83% in 2023. WUENIC incorporates the annual WHO and UNICEF estimates of national immunization coverage (WUENIC) and provides the world’s largest dataset on immunization coverage trends, based on country-reported data for 16 vaccines/antigens/doses across 195 WHO and UNICEF member states [46].

Furthermore, in contrast to other low- and middle-income countries, PCV-B in India has been consistently above the 80% mark since last year [47,48].

Figure 5 showcases the narrowing gap between the PCV B coverage and the proportion of the birth cohort to whom the PCV is available. As per the WUENIC data, the gap between PCV coverage and the cohort to whom the PCV is available has been consistently reducing since 2021.

#### 3.4.2. Potential Impact

The Integrated Global Action Plan for the Prevention and Control of Pneumonia and Diarrhea (GAPPD), launched by the WHO and UNICEF in 2009 and updated in 2013, lays out a comprehensive package of interventions aimed at ending preventable pneumonia and diarrhea-related child deaths by 2025. Each year, an analysis is made of the GAPPD scores, based upon 10 key indicators, to track global progress toward GAPPD targets. The increasing coverage of the PCV rollout has contributed to improvements in both the pneumonia and overall GAPPD scores for India [49,50] (Figure 6). As shown by the improvement of the GAPPD score from 2015 to 2024, it is evident that the introduction of the PCV in India has had a profound impact on child health, significantly reducing mortality among children under 5 years old. One of the most striking effects has been the reduction in pneumonia and diarrhea-related deaths, which declined by 54.39% between 2016 and 2024. The number of deaths from these causes fell from 260,990 in 2016 to 119,012 in 2024, indicating the vaccine’s effectiveness in preventing severe pneumococcal infections, a leading cause of childhood pneumonia.

Since both the Rotavirus vaccine and the PCV were introduced across India during this duration, we have looked at the mortality rates of pneumonia and diarrhea together. Upon examining the available data for childhood mortality due to pneumonia in India, there is a sharp drop between 2017 and 2023 (Figure 7) [3,51].

Furthermore, the overall under-5 mortality rate has also seen a notable decline, decreasing from 41.06 deaths per 1000 live births in 2016 to 29.06 deaths per 1000 live births in 2022. This significant reduction has contributed to saving approximately 334,800 lives annually in the under-5 age group, suggesting the critical contributory role of the PCV in strengthening child survival efforts. By contributing to the averting of an estimated 334,800 deaths in this age group, the vaccine has probably played a pivotal role in improving child health outcomes in India, reinforcing the importance of sustained immunization programs to further enhance child survival rates [52,53,54].

Future large-scale population-based surveillance studies would be required to quantify the impact of the PCV on the under-5 mortality rate and overall mortality and morbidity rates associated with pneumococcal diseases. This has been performed in other LMICs to establish the impact of PCV introduction and support its introduction in other countries [55,56,57].

#### 3.4.3. Product Switch

PCV13 (PREVNAR) was the first vaccine to be introduced in the UIP in India in 2017. This was followed by the introduction of PCV10 (PNEUMOSIL) in 2021, which was the first indigenous PCV. In 2024, another PCV product was licensed for use under the UIP-PCV14 rollout (PNEUBEVAX 14™). This is the second indigenous PCV product from a domestic manufacturer. Currently, two PCV products—PCV10 and PCV14—are available under the UIP. Both PCV10 and PCV14 offer defense against the most prevalent strains that cause pneumococcal illness [58,59]. PCV13 was phased out with the switch to newer PCV products available under the UIP. As is the case with a new vaccine introduction, the product switch involved regulatory approval, followed by transition planning for the new product. This included updating cold-chain storage, distribution networks, and healthcare worker training.

## 4. Discussion

The introduction of the pneumococcal conjugate vaccine (PCV) in India represents a significant milestone in public health. Its successful, phased rollout provides an opportunity to understand what worked, why it worked, and how.

Comparing PCV introduction in India with other large-scale vaccine rollouts, such as the introduction of the PCV in Latin America and the Caribbean (LAC) and other middle-income countries (MICs), helps to contextualize the steps taken to ensure a successful rollout.

Countries in LAC demonstrated that strong coordination between political and technical decision-makers was essential for a smooth introduction process, as was the case in India [60]. The critical decision-making process for PCV introduction in India aligned with the global trend of prioritizing high-burden areas initially [61].

One of the primary lessons from PCV introductions worldwide is the need for financial sustainability. MICs have faced challenges with vaccine affordability and limited external funding, highlighting the need for sustainable financing models [62] In Latin America, vaccine introduction faced barriers such as high costs and competing national health priorities [62,63]. In India, the affordability of PCVs was improved with the introduction of an indigenous vaccine (Pneumosil), which facilitated the nationwide scale-up [32].

In the Americas, proactive adjustments to the cold chain and logistical planning were instrumental in ensuring vaccine availability [60]. India’s phased approach to PCV introduction was successful, due to its structured implementation, but further analysis is needed to evaluate how logistical challenges, such as vaccine stockouts and distribution delays, were addressed. Experiences from fragile and humanitarian settings emphasize the importance of tailored approaches for vaccine implementation. Countries such as Chad, Guinea, Somalia, and South Sudan have faced numerous barriers, including low maternal education levels, climate risks, and inadequate surveillance, which impacted vaccine rollout [63]. India’s experience should be analyzed for similar challenges in underserved regions to improve access and uptake.

In comparison with other countries, the introduction and rapid expansion of the PCV rollout in India bring to the fore multiple significant lessons, as highlighted in the present study (Table 1). One of the major successes of the PCV rollout has been the ability to leverage the existing immunization infrastructure to ensure widespread coverage. The phased approach allowed for systematic implementation, and innovative tools such as the PCV Monitoring and Preparedness Tool (PROMPT) facilitated efficient monitoring despite the challenges posed by the COVID-19 pandemic. Additionally, hybrid training models incorporating digital platforms enabled timely capacity-building among healthcare workers, ensuring the vaccine’s smooth integration into the UIP.

While these achievements are commendable, the present study highlights areas requiring future research. A need to focus on evaluating the long-term effectiveness of PCVs in reducing under-5 mortality and the morbidity associated with pneumococcal diseases could be an important area for future studies. Such large-scale population-based surveillance is essential to quantify the impact of PCV introduction, as demonstrated in other low- and middle-income countries (LMICs) [4,64]. A study conducted in Kenya showed the effects of different PCV products (PCV10, PCV13, and the recently introduced PCV14) on the reduction of the pneumococcal disease burden. Similar assessments should be made to determine the most effective vaccine formulation for the Indian context [65]. This is especially important since, in other countries, a lack of disease burden surveillance has delayed vaccine decision-making [66,67].

Another crucial area for research is the effect of PCV introduction on antimicrobial resistance (AMR). Several studies have shown that PCVs contribute to reducing antibiotic use and AMR by preventing bacterial infections that would otherwise require antibiotic treatment [68,69]. Evaluating the impact of PCVs on AMR in India could provide valuable insights into broader public health benefits beyond direct disease prevention. Furthermore, serotype replacement and the emergence of non-vaccine serotypes remain concerns following PCV introduction. The continuous monitoring of pneumococcal serotype distribution and invasive pneumococcal disease (IPD) patterns will help guide vaccine policy decisions and the potential need for next-generation PCVs [70,71]. India’s immunization program could also invest further in robust epidemiological surveillance to track the PCV rollout’s long-term impact on resistance trends [72].

Additionally, cost-effectiveness studies assessing the economic benefits of PCV introduction should be conducted to provide policymakers with evidence to support sustained funding and vaccine procurement decisions [73,74]. While PCV introduction in the UIP has significantly increased its accessibility, future research should explore equity in vaccine coverage, particularly among marginalized and underserved populations. Understanding the barriers to vaccine uptake and designing targeted interventions will help ensure equitable immunization access across all socioeconomic groups [74].

Furthermore, the role of PCV in preventing pneumonia-related hospitalizations and outpatient visits warrants detailed investigation. Studies assessing healthcare utilization trends pre- and post-PCV introduction can provide a clearer picture of the vaccine’s impact on the overall healthcare burden. Such data would be instrumental in refining health system strategies and resource allocation to maximize public health benefits. While India’s PCV introduction journey has been a remarkable public health achievement, continued research is essential to maximize its impact.

## 5. Conclusions

Pneumococcal pneumonia is one of the leading causes of lower respiratory diseases and death among children under 5 years of age in India. To reduce the burden of pneumonia, the MoHFW decided to launch the PCV in a phased manner. With the introduction of the PCV in UIP, millions of children receive protection against the leading cause of pneumonia. It was introduced initially in five high-burden states, following which pan-India expansion was completed in 2021. The COVID-19 pandemic coinciding with the PCV expansion posed multiple challenges, necessitating innovative approaches to ensure the smooth rollout of the PCV across the remaining 31 states/UTs of India. The careful and intricate planning and meticulous execution by the MoHFW, with the support of development partners and other stakeholders, allowed a successful rollout of PCV in tandem with the world’s largest COVID-19 vaccination drive.

Since the pan-India expansion of the PCV rollout in 2021, its national-level coverage has steadily increased. PCV booster coverage—an indicator of the completion of PCV doses—has been consistently above 80% in India over the past year, outperforming other low- and middle-income countries. The WUENIC data highlights this upward trend, with the PCV contributing positively to the WHO and UNICEF’s Integrated Global Action Plan for the Prevention and Control of Pneumonia and Diarrhea (GAPPD) scores. However, future population-based studies will be required to document the PCV’s impact on reducing child mortality and morbidity in India.

## Figures and Tables

**Figure 1 vaccines-13-00432-f001:**
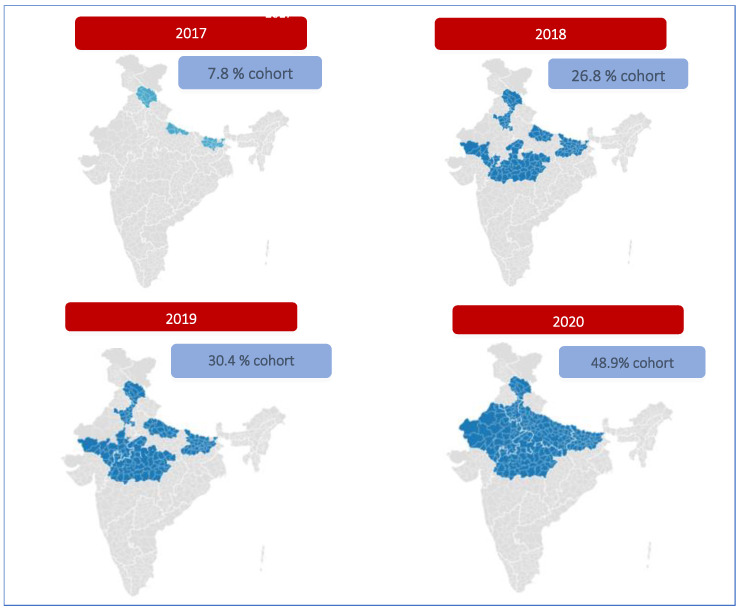
Geographic extent of PCV introduction according to district, 2017–2020.

**Figure 2 vaccines-13-00432-f002:**
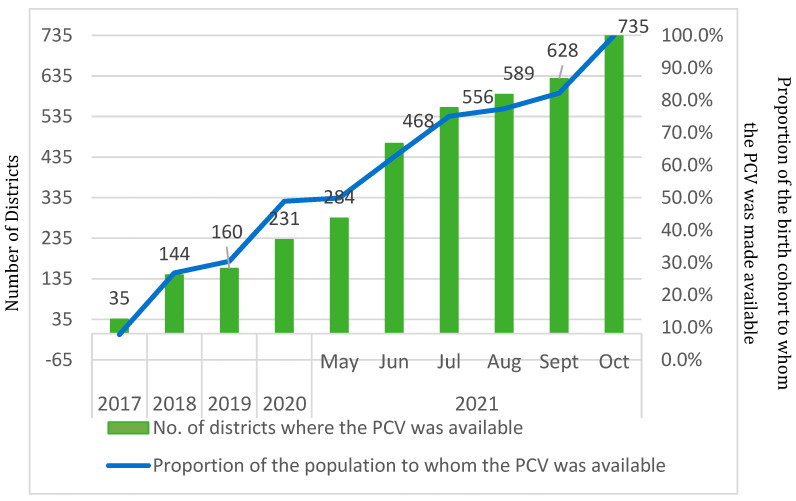
Snapshot of PCV expansion in India, showing the proportion of the birth cohort to whom the PCV was made available.

**Figure 3 vaccines-13-00432-f003:**
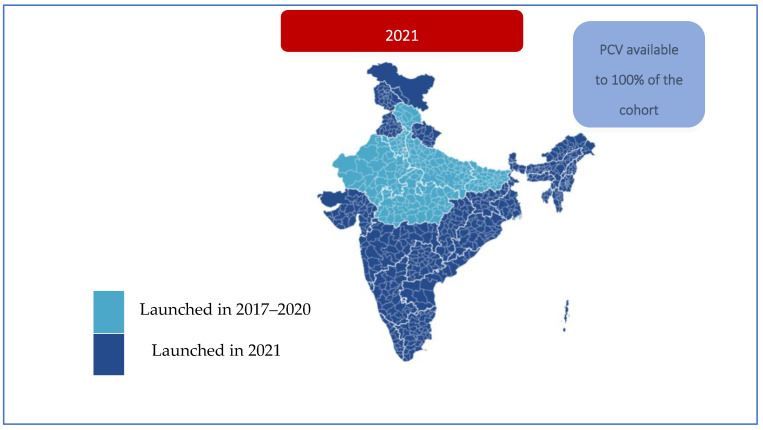
Pan-India PCV expansion.

**Figure 4 vaccines-13-00432-f004:**
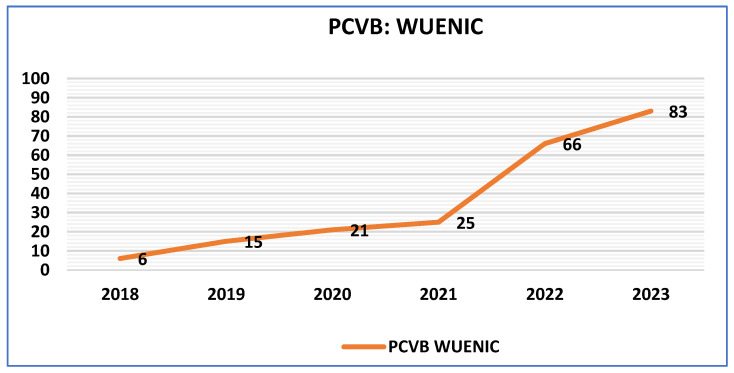
PCV B coverage, reported as per WUENIC in 2024.

**Figure 5 vaccines-13-00432-f005:**
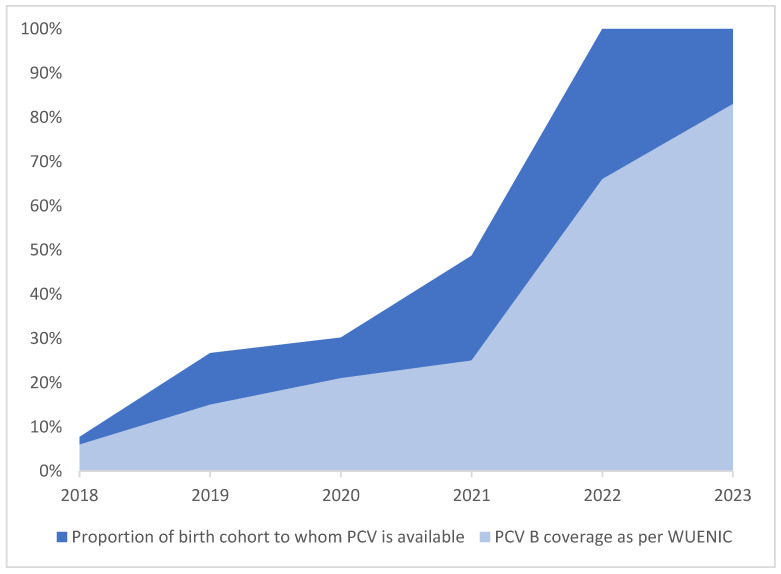
The PCV booster coverage (WUENIC) trend vis-à-vis the phased introduction.

**Figure 6 vaccines-13-00432-f006:**
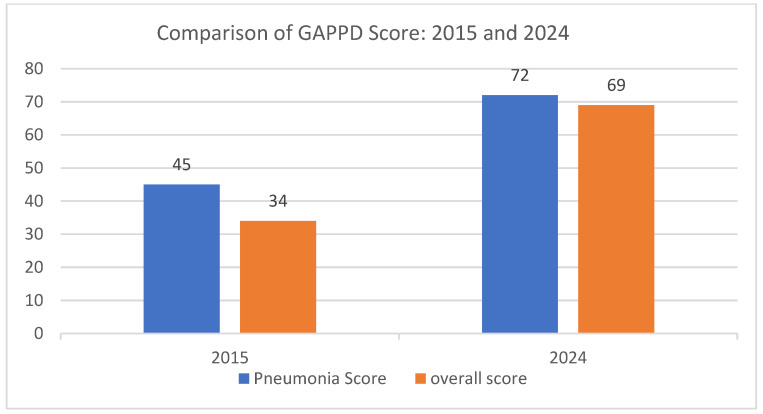
Comparison of India’s GAPPD scores between 2015 and 2024.

**Figure 7 vaccines-13-00432-f007:**
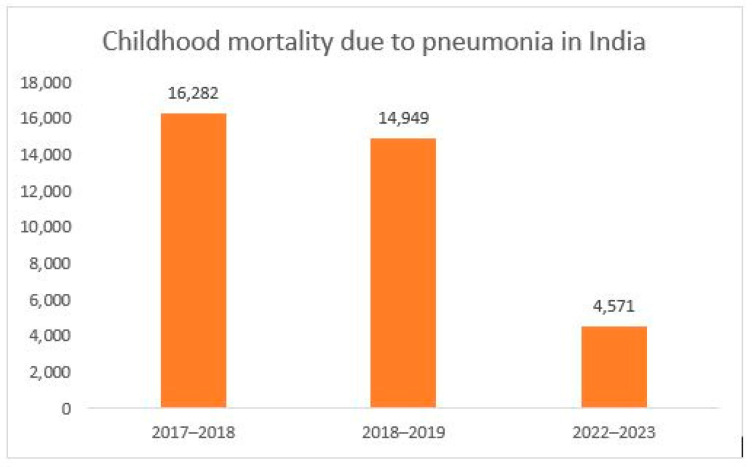
Pneumonia mortality among children under 5 years old in India over the years.

**Table 1 vaccines-13-00432-t001:** Adaptive measures taken during the COVID-19 pandemic for PCV expansion in India.

Component	Challenges Due to the COVID-19 Pandemic	Adaptive Measures Adopted
**Preparedness** **Assessment**	- Manual documentation and supervisory visits were restricted, due to lockdown and travel limitations.	- Developed PROMPT (PCV Rollout Monitoring and Preparedness Tool) for digital assessment. - Real-time monitoring and an interactive dashboard enabled efficient tracking.
**Training**	- Large-scale in-person training was not feasible. - Social distancing and travel restrictions affected training sessions.	- Hybrid training model (online + classroom). - Development of FAQs, operational guidelines, and five animated videos in 13 languages. - Dissemination through WhatsApp for wider reach.
**Community Mobilization**	- Restrictions on public gatherings limited traditional community engagement efforts.	- Use of traditional media (radio, posters, hoardings, and leaflets) and digital media (WhatsApp videos and social media campaigns). - Engagement of celebrities and public figures for promotion.
**Logistics and Cold-Chain Space Management**	- Need for adequate storage space amidst the COVID-19 vaccine rollout. - Risk of equipment failure in some areas.	- Cold-chain space assessment and augmentation before PCV expansion. - Use of the eVIN (Electronic Vaccine Intelligence Network) for real-time supply chain monitoring. - Repair/replacement of non-functional ILRs and deep freezers.
**Monitoring and Supervision**	- Difficulty in tracking real-time implementation, due to movement restrictions.	- Rapid field monitoring using standardized data collection formats. - Mobile-based data collection and visualization for easy monitoring.

## Data Availability

Not applicable.

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
