# Peer review of "Chronicling the Journey of Pneumococcal Conjugate Vaccine Introduction in India"

_vaccines, 2025, doi:10.3390/vaccines13040432_

Round 1

Reviewer 1 Report

Comments and Suggestions for Authors

The manuscript describes the introduction of conjugate pneumococcal vaccines in India., which as expanded nationwide in 2021, with increased coverage. The subject is highly relevant as a narrative of vaccine introduction, but there are some concerns, which I have pointed out below:
1.    The abstract needs more concrete data. The percentage of coverage is of vital importance; the authors just mention a “steady increase”, but do not provide the numbers.
2.    Lines 91-100 – since there are different types of PCV, it is important to identify which formulation was evaluated in each study.
3.    Line 114, GOI is all capital letters. Line 55 reads GoI. Homogenize.
4.    3.1.2 when describing the decision to use PCV13, it would be interesting to know the predicted efficacy of this formulation in the Indian population, as well as the serotype distribution in that country at the time of introduction.
5.    Line 207, remove extra space. Same in lines 299, 321.
6.    Figure 4 should be better explored in the text, indicating the increase in vaccine uptake with numbers. 
7.    Figure 5 is hard to read, since the two shades of blue are too similar. 
8.    Lines 301-302 – This needs to be better discussed. It is important to compare the potential impact of each formulation based on the pneumococcal serotype prevalence in India

Author Response

  1. The abstract needs more concrete data. The percentage of coverage is of vital importance; the authors just mention a “steady increase”, but do not provide the numbers.

Response: We appreciate the reviewer's feedback on the abstract. We have provided the numbers in the abstract in the revised version of the manuscript.

  1. Lines 91-100 – since there are different types of PCV, it is important to identify which formulation was evaluated in each study.

 Response: Thank you for your feedback. The different types of PCV according to the studies have been identified and mentioned in the revised manuscript.

  1. Line 114, GOI is all capital letters. Line 55 reads GoI. Homogenize.

Response: Thank you for pointing this out. Line 55 and 55 have been homogenized as GoI.

  1. 3.1.2 when describing the decision to use PCV13, it would be interesting to know the predicted efficacy of this formulation in the Indian population, as well as the serotype distribution in that country at the time of introduction.

Response: Thank you for your feedback. The recommendation of NTAGI was based on global experience on the efficacy of PCV.

  1. Line 207, remove extra space. Same in lines 299, 321.

Response: Thank you for pointing this out. Extra space has been removed.

  1. Figure 4 should be better explored in the text, indicating the increase in vaccine uptake with numbers.

Response: Thank you for your feedback. Figure 4 is explained in the text in revised manuscript.

  1. Figure 5 is hard to read, since the two shades of blue are too similar.

Response: Thank you for pointing this out. The shades in figure 5 have been changed.

  1. Lines 301-302 – This needs to be better discussed. It is important to compare the potential impact of each formulation based on the pneumococcal serotype prevalence in India

Response: Thank you for your feedback. We have tried to address this concern in section 3.4.2

Reviewer 2 Report

Comments and Suggestions for Authors

Many thanks for your submission, this is a great piece of work highlighting the need for pneumococcal vaccination however there are multiple spelling and grammatical errors which need to be corrected; 

"Pneumococcal Conjugate Vaccines (PCV) was introduced" - "Pneumococcal Conjugate Vaccines (PCVs) were introduced"

"a major initiative by Ministry of Health" - "a major initiative by the Ministry of Health"

"Despite challenges posed by COVID-19 pandemic" - "Despite challenges posed by the COVID-19 pandemic"

"This narrative review highlights significant decisions, dedication and coordinated efforts" - "This narrative review highlights significant decisions, dedication, and coordinated efforts"

"A number of studies have brought to fore" - "A number of studies have brought to the fore"

"Considering the large burden of disease and the vaccine preventable nature of pneumococcal diseases" - "Considering the large burden of disease and the vaccine-preventable nature of pneumococcal diseases"

"the Government of India guaranteed fair access of PCV" - "the Government of India guaranteed fair access to PCV"

"multiple processes spread over 4 years, multiple phases and 36 states/UTs" - "multiple processes spread over four years, multiple phases, and 36 states/UTs"

"necessitating the development of various innovative tools, approaches, and processes to successfully roll out the vaccine"

"A number studies showcased the immunogenicity and safety" - "A number of studies showcased the immunogenicity and safety"

You need to use MoHFW consistently, you have written it out multiple times, abbreviate it once at the start then use MoHFW throughout. 

the references are very inconsistent e.g. 42 and 43 - the spacing is incorrect - please recheck all references to make sure they comply with the journals requirements. 

the methodology needs further work - did you search from inception to a current date? there needs to be a reference point of when the searches were conducted with dates. did you use any other resources such as embase or medline ? 

how did you overcome risk of bias in this - was this considered ? needs to be in the methodology 

paragraph 3.1 needs a reference 

IEC on line 152 needs to clarification on what IEC stands for 

line 174 - i would suggest describing PCV10 as a conjugate vaccine, to the wider audience, i am not sure 'indigenous' would be translatable 

in the conclusion - i would suggest these changes

"bring down the burden" - "reduce the burden" 

"vaccination drive in the world" - "world’s largest COVID-19 vaccination drive"

in the learnings section - i would change this to 'lessons' 

significant learnings" - "significant lessons" 

Comments on the Quality of English Language

I would advise getting a native speaker to read through this manuscript, i suspect they will change the way its written, it reads ok but there is scope for improvement on the flow of the article. 

Author Response

  1. "Pneumococcal Conjugate Vaccines (PCV) was introduced" - "Pneumococcal Conjugate Vaccines (PCVs) were introduced"

Response: Thank you for pointing this out. Spelling and grammatical errors has been corrected.

  1. "a major initiative by Ministry of Health" - "a major initiative by the Ministry of Health"

Response: Thank you for your comment. This has been addressed.

  1. "Despite challenges posed by COVID-19 pandemic" - "Despite challenges posed by the COVID-19 pandemic"

Response: Thank you. This has been addressed.

  1. "This narrative review highlights significant decisions, dedication and coordinated efforts" - "This narrative review highlights significant decisions, dedication, and coordinated efforts"

Response: Thank you for your comment. This has been addressed

  1. "A number of studies have brought to fore" - "A number of studies have brought to the fore"

Response: Thank you for your comment. We have made corrections.

  1. "Considering the large burden of disease and the vaccine preventable nature of pneumococcal diseases" - "Considering the large burden of disease and the vaccine-preventable nature of pneumococcal diseases"

Response: Thank you for your comment. This has been addressed.

  1. "The Government of India guaranteed fair access of PCV" - "the Government of India guaranteed fair access to PCV"

Response: Thank you. We have made the required changes.

  1. "Multiple processes spread over 4 years, multiple phases and 36 states/UTs" - "multiple processes spread over four years, multiple phases, and 36 states/UTs"

Response: Thank you for your comment. This has been addressed

  1. "Necessitating the development of various innovative tools, approaches, and processes to successfully roll out the vaccine"

Response: Thank you for your comment. This has been addressed

  1. "A number studies showcased the immunogenicity and safety" - "A number of studies showcased the immunogenicity and safety"

Response: Thank you for your comment. This has been addressed.

  1. You need to use MoHFW consistently, you have written it out multiple times, abbreviate it once at the start then use MoHFW throughout.

Response: Thank you for your comment. This has been addressed.

  1. the references are very inconsistent e.g. 42 and 43 - the spacing is incorrect - please recheck all references to make sure they comply with the journal’s requirements.

Response: Thank you for pointing this out. The spacing in reference 42 and 43 have been corrected

  1. The methodology needs further work - did you search from inception to a current date? there needs to be a reference point of when the searches were conducted with dates. did you use any other resources such as embase or medline?

Response: Thank you for your feedback. We have added details for the same under the methodology section.

  1. How did you overcome risk of bias in this - was this considered? needs to be in the methodology?

Response: We have addressed this question. We have added the details under the methodology section.

  1. Paragraph 3.1 needs a reference

Response: Thank you for pointing this out. Reference has been added in the revised version.

  1. IEC on line 152 needs to clarification on what IEC stands for

Response: Thank you for pointing this out. Full form of IEC has been mentioned.

  1. Line 174 - i would suggest describing PCV10 as a conjugate vaccine, to the wider audience, i am not sure 'indigenous' would be translatable

Response: Thank you for pointing this out. This line has been described as “Pneumosil, an indigenous vaccine developed by Serum Institute of India, which is a 10-valent PCV product” in the revised section.

  1. in the conclusion - i would suggest these changes

"bring down the burden" - "reduce the burden"

"vaccination drive in the world" - "world’s largest COVID-19 vaccination drive"

in the learnings section - i would change this to 'lessons'

significant learnings" - "significant lessons"

Response: Thank you for pointing this out. These have been addressed.

Reviewer 3 Report

Comments and Suggestions for Authors

The manuscript presents a comprehensive overview of the phased introduction and nationwide expansion of Pneumococcal Conjugate Vaccine (PCV) in India. While the study is relevant and contributes valuable insights into large-scale immunization efforts, several key issues need to be addressed before publication. The primary concerns relate to the methodology, clarity and depth of process descriptions, and integration of critical analysis. Additionally, the manuscript would benefit from a more structured approach to discussing challenges, limitations, and future implications. Here are some major suggestions. 

1. Methodology needs greater detail and rigor. The methodology section lacks sufficient details regarding criteria for literature selection, inclusion/exclusion parameters, and data synthesis approaches. Given that this is a desk review, the process of gathering and validating data from different sources should be made explicit. Also, the manuscript states that both published and unpublished literature were reviewed, but it does not describe what grey literature was assessed for quality and reliability. 

2. Overly descriptive narrative without analytical depth. While the paper successfully chronicles the introduction of PCV, it does so in an overly linear and descriptive manner. A stronger analytical lens is needed to explore why certain strategies were effective, what lessons were learned, and how these insights contribute to global immunization literature. Consider adding comparative analysis with other large-scale vaccine introductions to provide context and demonstrate broader applicability of findings. Also, the discussion should reflect on gaps in data, policy decisions that shaped outcomes, and areas where improvements could have been made.

3. Although the paper acknowledges the challenges posed by COVID-19, it does not sufficiently analyze how these were mitigated and what adaptive strategies were employed. The section on cold chain management and logistics is too high-level. It should delve into specific bottlenecks encountered, their impact on vaccine distribution, and how they were overcome. How they switch the product should also be introduced in more detail. Now, most of these important parts didn't include too much information on how they make it possible.

4. The manuscript discusses the increasing PCV coverage and references WHO/UNICEF estimates, but it lacks a comprehensive presentation of time-series data, impact assessments, and statistical trends. If possible, include coverage rates by region, vaccination dropout rates, or comparative data on pre- and post-introduction pneumonia mortality rates. 

5. The manuscript briefly mentions the importance of further research but does not specify what key questions remain unanswered.

Author Response

  1. Methodology needs greater detail and rigor. The methodology section lacks sufficient details regarding criteria for literature selection, inclusion/exclusion parameters, and data synthesis approaches. Given that this is a desk review, the process of gathering and validating data from different sources should be made explicit. Also, the manuscript states that both published and unpublished literature were reviewed, but it does not describe what grey literature was assessed for quality and reliability.

Response: Thank you for this comment. We have tried to address the same with additions to the existing methodology. The following were the basis of relying heavily on desk reviews without a systematic approach:

  • The present manuscript relies on incorporation of both published and unpublished literature, the methodology ensures a comprehensive understanding of the topic, capturing insights that may not be available in indexed journals. This approach helps include grey literature, policy briefs, and institutional reports that are often overlooked but highly relevant.
  • A strict systematic approach may exclude valuable sources due to rigid inclusion/exclusion criteria. The current desk review method allows for adaptability in identifying emerging trends and diverse perspectives. This is particularly useful in rapidly evolving fields, such as vaccines, where new findings frequently emerge outside conventional peer-reviewed sources.
  • Instead of relying solely on predefined inclusion/exclusion criteria, expert judgment is applied to assess the relevance and credibility of sources. This allows for a more nuanced interpretation of data rather than mechanical filtering.

.

  1. Overly descriptive narrative without analytical depth. While the paper successfully chronicles the introduction of PCV, it does so in an overly linear and descriptive manner. A stronger analytical lens is needed to explore why certain strategies were effective, what lessons were learned, and how these insights contribute to global immunization literature. Consider adding comparative analysis with other large-scale vaccine introductions to provide context and demonstrate broader applicability of findings. Also, the discussion should reflect on gaps in data, policy decisions that shaped outcomes, and areas where improvements could have been made.

Response: This is very helpful. We have addressed this and added a comparative with other countries in our discussion section.

  1. Although the paper acknowledges the challenges posed by COVID-19, it does not sufficiently analyze how these were mitigated and what adaptive strategies were employed. The section on cold chain management and logistics is too high-level. It should delve into specific bottlenecks encountered, their impact on vaccine distribution, and how they were overcome. How they switch the product should also be introduced in more detail. Now, most of these important parts didn't include too much information on how they make it possible.

Response: Thank you for your comment. A detailing of all aspects of new vaccine introduction was beyond the scope of the present manuscript. In response to your comment we have added details on cold chain management and logistics section. Meanwhile, we have summarized the adaptive measures in a table.

  1. The manuscript discusses the increasing PCV coverage and references WHO/UNICEF estimates, but it lacks a comprehensive presentation of time-series data, impact assessments, and statistical trends. If possible, include coverage rates by region, vaccination dropout rates, or comparative data on pre- and post-introduction pneumonia mortality rates.

Response: Thank you for your comment. A detailing of all aspects of new vaccine introduction was beyond the scope of the present manuscript. In response to your comment, we have added details on cold chain management and logistics section.

  1. The manuscript briefly mentions the importance of further research but does not specify what key questions remain unanswered.

Response: As highlighted in the section- Lessons and Future Research Implications, the following are the key research areas which remain unanswered.

  • Impact on Antimicrobial Resistance (AMR) – While existing evidence suggests PCVs reduce antibiotic use and AMR, India-specific studies are needed to evaluate these effects in the local epidemiological context.

  • Serotype Replacement and Disease Patterns – Continuous monitoring of pneumococcal serotype distribution and invasive pneumococcal disease (IPD) patterns will help assess whether non-vaccine serotypes emerge and influence disease burden.

  • Cost-effectiveness Analysis – Evaluating the economic benefits of PCV introduction will provide policymakers with data to support sustained funding and procurement decisions.

  • Vaccine Equity and Coverage – Research should explore disparities in vaccine access among marginalized and underserved populations, identifying barriers to uptake and informing targeted interventions.

  • Healthcare Utilization Trends – Assessing pneumonia-related hospitalizations and outpatient visits before and after PCV introduction will provide insights into the vaccine’s broader impact on the healthcare system.

Reviewer 4 Report

Comments and Suggestions for Authors

The authors undertook a very comprehensive assessment of the nationwide roll-out of conjugate pneumococcal vaccination in India.

I have only minor but essential amendments:

  1. The abstract needs to contain numerical figures regarding PCV coverage and WUENIC.
  2. On page 2, line 44 the authors need to delete "(Pneumococcal pneumoniae)" as this name does not exist.
  3. On page 3, line 103: The authors need to delete "or without other infiltrates" and the "+/- OI" as this part of the sentence is unclear. Where are the infiltrates if not already covered by the term pneumonia?
  4. In the method section the authors need to define and explain the meaning of WUENIC and the procedures involved.
  5. Page 8 and 9:The authors need to explain the way mortality was measured and only focussing on mortality figures for pneumonia because diarrhoea deaths are not caused by Streptococcus pneumoniae. The authors need to just report pneumonia morbidity and mortality before and after vaccine exposure.
  6. Page 9: From line 304 onwards: The authors need to provide a paragraph on the potential impact on bacterial meningitis aetiology and epidemiology.

Author Response

1. The abstract needs to contain numerical figures regarding PCV coverage and WUENIC.

Response: Thank you for your feedback. We have mentioned the numerical figures regarding PCV coverage and WUENIC as per the suggestion.

2. On page 2, line 44 the authors need to delete "(Pneumococcal pneumoniae)" as this name does not exist.

Response: We thank the reviewer for pointing this out. We have removed this section.

3. On page 3, line 103: The authors need to delete "or without other infiltrates" and the "+/- OI" as this part of the sentence is unclear. Where are the infiltrates if not already covered by the term pneumonia?

Response: We agree with the reviewer that this part looks unclear. We have deleted this part.

4. In the method section the authors need to define and explain the meaning of WUENIC and the procedures involved.

Response: Thank you for this comment. The definition and explanation of WUENIC have been provided in abstract and in section 3.4.1.

5. Page 8 and 9: The authors need to explain the way mortality was measured and only focussing on mortality figures for pneumonia because diarrhoea deaths are not caused by Streptococcus pneumoniae. The authors need to just report pneumonia morbidity and mortality before and after vaccine exposure.

Response: Since both Rotavirus vaccine and PCV were introduced pan India during this du-ration, we have looked at the mortality rates of pneumonia and diarrheoa together. Although, on examining the available data for Childhood Mortality due to Pneumonia, there is a sharp drop between 2017 and 2023. We have added the graph for the same to the manuscript.

6. Page 9: From line 304 onwards: The authors need to provide a paragraph on the potential impact on bacterial meningitis aetiology and epidemiology.

Response: Thank you for this comment. We have mentioned the potential impact on bacterial meningitis aetiology and epidemiology in section 3.1.1.

Reviewer 5 Report

Comments and Suggestions for Authors

The manuscript presents a comprehensive narrative review detailing the phased introduction of the Pneumococcal Conjugate Vaccine (PCV) in India. The authors have provided an extensive discussion on the implementation process, challenges encountered, and outcomes achieved. However, the manuscript requires significant revisions to enhance its scientific rigor, clarity, and coherence. Below are major concerns and recommendations for improvement.

  1. The manuscript is lengthy and somewhat repetitive in its descriptions of the phased rollout. Consider streamlining content to avoid redundancy
  2. The methodology relies heavily on desk reviews of published and unpublished literature. However, it lacks a systematic approach and details on search strategy, inclusion/exclusion criteria, and quality assessment of the included studies

  3. While the manuscript provides an extensive narrative, it lacks quantitative analyses to support its claims. The authors should consider incorporating more statistical data on vaccination rates, impact on pneumococcal disease burden, and regional variations

  4. Graphs and figures are presented but require more explanation within the text. Ensure all data sources are cited appropriately

  5. A comparison with other countries’ PCV implementation strategies would provide additional context.

Author Response

  1. The manuscript is lengthy and somewhat repetitive in its descriptions of the phased rollout. Consider streamlining content to avoid redundancy.

Response: Thank you for your feedback. We agree that the manuscript is lengthy. We have removed the sections which we thought we repetitive.

  1. The methodology relies heavily on desk reviews of published and unpublished literature. However, it lacks a systematic approach and details on search strategy, inclusion/exclusion criteria, and quality assessment of the included studies.

Response: Thank you for this comment. We have tried to address the same with additions to the existing methodology. The following were the basis of relying heavily on desk reviews without a systematic approach:

  • The present manuscript relies on incorporation of both published and unpublished literature, the methodology ensures a comprehensive understanding of the topic, capturing insights that may not be available in indexed journals. This approach helps include grey literature, policy briefs, and institutional reports that are often overlooked but highly relevant.
  • A strict systematic approach may exclude valuable sources due to rigid inclusion/exclusion criteria. The current desk review method allows for adaptability in identifying emerging trends and diverse perspectives. This is particularly useful in rapidly evolving fields, such as vaccines, where new findings frequently emerge outside conventional peer-reviewed sources.
  • Instead of relying solely on predefined inclusion/exclusion criteria, expert judgment is applied to assess the relevance and credibility of sources. This allows for a more nuanced interpretation of data rather than mechanical filtering.
  1. While the manuscript provides an extensive narrative, it lacks quantitative analyses to support its claims. The authors should consider incorporating more statistical data on vaccination rates, impact on pneumococcal disease burden, and regional variations.

Response: The manuscript’s narrative approach has been utilized as it provides a comprehensive thematic synthesis, offering valuable insights into policy implications, vaccine adoption challenges, and regional disparities that raw statistics alone may not capture. Since vaccination rates could have large variations across regions and the present manuscript as dealt with India as a whole, analysis at regional level has been omitted. Additionally, undertaking quantitative statistical analyses for pneumococcal disease burden and PCV coverages is beyond the purview of the present manuscript which primarily aims to provide a narrative understanding of the subject, making it well-suited for policy discussions and broader public health considerations.

  1. Graphs and figures are presented but require more explanation within the text. Ensure all data sources are cited appropriately.

Response: Thank you for your feedback. All the graphs and figures have been explained within the text and all the citations are done properly.

  1. A comparison with other countries’ PCV implementation strategies would provide additional context.

Response: Thank you for this comment. We have added the comparison and discussion around it in the discussion section.

Round 2

Reviewer 1 Report

Comments and Suggestions for Authors

All my questions have been adressed properly.

Author Response

Response to reviewer 1

All my questions have been addressed properly.

Thank you so much for your feedback. We really appreciate your review. 

Reviewer 2 Report

Comments and Suggestions for Authors

thank you for the corrections - the manuscript is sutiable for publication. 

Author Response

Response to reviewer 2

Thank you for the corrections - the manuscript is suitable for publication. 

Thank you for your valuable feedback.

Reviewer 3 Report

Comments and Suggestions for Authors

Thank the authors for all the revisions. Here are two suggestions to consider.

  1. The discussion part is somewhat limited, followed by more contests in the lessons and future research implications. It would be better to combine these two parts to make it easier to follow and more logical for a broader audience.
  2. The maps in several figures are not very clear, and maps are not suggested to be used in many scientific papers due to multiple underline reasons. It would be better for the authors to use other ways to show these parts of the results; tables or a few sentences would be more preferable. 

Author Response

Response to reviewer 3

Thank the authors for all the revisions. Here are two suggestions to consider.

  1. The discussion part is somewhat limited, followed by more contests in the lessons and future research implications. It would be better to combine these two parts to make it easier to follow and more logical for a broader audience.

Thank you for your feedback. We have combined the discussion section and lessons and future research implications section.

  1. The maps in several figures are not very clear, and maps are not suggested to be used in many scientific papers due to multiple underline reasons. It would be better for the authors to use other ways to show these parts of the results; tables or a few sentences would be more preferable. 

Thank you for your feedback. We have included the maps to show geographical representations of PCV introduction which happened phase wise from 2017-2020 followed by nation-wide introduction by 2021.

Reviewer 5 Report

Comments and Suggestions for Authors

The authors response well.

Author Response

Response to reviewer 5

The authors response well.

Thank you for your feedback. We really appreciate it.